# The physical activity and sedentary behavior among pregnant women in Macao: A cross-sectional study

Ka Chon Mok[1], Ming Liu[2], Xin Wang ID [2]*

1 Faculty of Health Sciences and Sports, Macao Polytechnic University, Macao, China, 2 Peking University Health Science Center, Macao Polytechnic University Nursing Academy, Macao Polytechnic University, Macao, China

* amywang@mpu.edu.mo

**Data Availability Statement:** All relevant data are within the manuscript and its Supporting Information files.

**Funding:** This work was supported by Macao Polytechnic University [grant number RP/AE-02/

## Abstract

### Objective

The current investigation sought to elucidate the prevalence and contributing factors of sedentary behavior among pregnant women in Macao, a densely populated region characterized by a distinctive fusion of Eastern and Western cultures and a thriving global economy.

### Methods

Through a cross-sectional study design, a total of 306 expectant mothers were recruited via various social media platforms and completed a sociodemographic questionnaire alongside the Chinese version of the Pregnancy Physical Activity Questionnaire.

### Results

The findings revealed that sedentary activities accounted for a relatively small proportion (7.8%) of the participants' total activity energy expenditure. Interestingly, employment status emerged as a significant determinant, with employed pregnant women exhibiting a 57.9% lower risk of being sedentary compared to their unemployed counterparts. Moreover, multiparous women (those with two or more children) were approximately 9 times more likely to meet moderate-intensity activity standards than nulliparous women.

### Conclusion

These insights highlight the importance of tailoring physical activity interventions to address the specific needs and challenges faced by primiparous women and those who are unemployed during pregnancy, with a view to enhancing education on the potential hazards associated with sedentary habits and promoting active lifestyles within this unique sociocultural context.

2022] The funders had no role in study design, data collection and analysis, decision to publish, or preparation of the manuscript.

**Competing interests:** The authors have declared that no competing interests exist.

## 1. Introduction

Sedentary behavior (SB) refers to any waking behavior characterized by an energy expenditure ≤1.5 metabolic equivalents (METs) while in a sitting, reclining or lying posture [1]. Prolonged sedentary time among adults has been associated with increased risks of various chronic diseases. For pregnant women, inadequate physical activity (PA) or prolonged sedentary time can increase the risk of various pregnancy-related complications, such as obesity, gestational hypertension or gestational diabetes [2, 3]. These health problems may also exacerbate the risk of adverse outcomes in the newborn, posing threats to maternal and infant health [3]. Recent research has also shown that prolonged sitting time is associated with an elevated risk of all-cause mortality, suggesting SB as a risk factor for chronic diseases and premature death. Globally, it is estimated that around 3 million deaths per year are attributable to physical inactivity [1].

In 2020, the World Health Organization (WHO) updated its guidelines on lifetime PA, which for the first time included specific recommendations on PA during pregnancy and the postpartum period. These guidelines, based on the best available evidence, recommend that pregnant women engage in at least 150 minutes per week of moderate-intensity aerobic activity unless contraindicated [4]. Other studies have also documented additional health benefits of increased PA during pregnancy, including reduced risk of gestational diabetes [5], preterm birth [6], cesarean delivery [7], cardiovascular complications [8], rectus abdominis diastasis [9] and antenatal and postpartum depression [10, 11]. However, research indicates that the proportion of pregnant women meeting international PA guidelines remains low, both in developed and developing countries [12–14].

Numerous factors have been reported to contribute to the high risk of physical inactivity or decreased PA during pregnancy, including fatigue, lack of time, pregnancy-related discomforts [15, 16], fear of harming the fetus [17, 18], lack of knowledge [19], insufficient advice/information, and lack of social support [20].

The relationship between maternal age and physical activity during pregnancy is inconsistent. While some studies indicate that older pregnant women are more likely to meet recommended activity standards[21], others report no significant association between maternal age and physical activity levels [22]. The influence of previous exercise habits on PA during pregnancy is also inconclusive, with some studies supporting the positive effect of prior exercise [14], while others have found contradictory results [23]. Socioeconomic factors such as education level and household income have also yielded mixed conclusions, with some research suggesting a positive correlation between higher education, higher income and PA levels during pregnancy [24], while other studies have not found such associations [14, 22, 23, 25]. In summary, the factors influencing PA during pregnancy remain not fully clear, and further in-depth research is needed to draw more consistent conclusions.

A cohort study conducted across 15 provinces in China revealed that the levels of PA among pregnant women are relatively low, only about one-third of pregnant women achieved sufficient PA across all three trimesters, and nearly 10% experienced persistent insufficient PA [22]. Currently, developed countries such as the United States, Canada, Australia, and the United Kingdom have published various versions of guidelines and recommendations for PA during pregnancy. In contrast, China has only a limited number of studies analyzing foreign pregnancy exercise guidelines, empirical benefits of PA during pregnancy, and the significant inadequacy of PA among domestic pregnant women. There is a noticeable lack of expert consensus or recommendations regarding exercise during pregnancy in China [26].

The issue of PA and SB among pregnant women in Macao remains unexplored. Given Macao's unique blend of Chinese and Western cultures and its high per capita GDP [27],

regional culture and income may impact PA levels. This study aims to fill this research gap, offering insights to enhance PA habits among pregnant women and support their overall health and well-being in the region.

## 2. Materials and methods

### 2.1 Design and setting

The present study employed a cross-sectional, descriptive quantitative research design. A convenience sampling approach was applied, with the primary recruitment site being various online social media platforms hosting Macao-based pregnancy groups. This includes 12 WeChat groups corresponding to the respective months of expected delivery, representing a large-scale mutual support network for expectant mothers in Macao, with a total of 4,500 members. From October 1, 2023, to February 29, 2024, the Questionnaire, QR codes or weblinks were posted in these groups, allowing eligible pregnant women to complete the survey online.

### 2.2 Participants

Official statistical data on the total number of pregnant women in Macao are currently unavailable. Therefore, the present study referenced the number of newborn infants as a proxy measure. Assuming that each pregnant woman delivers an infant approximately every nine months, and considering the variability in pregnancy duration, it is essential to calculate the total number of newborns overborn in a specified period frame (such as one year) to provide a more accurate estimate of the number of pregnant women. When this research was initiated, the statistical data for newborn infants in 2023 had not yet been published. Consequently, the study relied on the 2022 statistical data provided by the Macao Social Welfare Bureau, which reported 4,344 newborn infants in that year [28]. Based on this information, the total estimated population of pregnant women in Macao during the 2022–2023 period was approximately 4,340, and this figure was used as the basis for the study's target population.

The inclusion criteria comprised individuals who were: 1) aged 18 or above and held either permanent or non-permanent resident status in Macao; 2) confirmed their pregnancy between October 2023 and February 2024; 3) proficient in reading and writing Chinese and capable of completing the online questionnaire using electronic devices. Exclusions encompassed participants: 1) with significant chronic conditions that could affect pregnancy-related PA, like musculoskeletal disorders; 2) with multiple gestations; 3) whose questionnaires had duplicate IP addresses.

The sample size was calculated using a single population proportion formula, based on a 95% confidence interval and a 5% margin of error. Previous research [29] reported a 25.61% prevalence of sedentary lifestyle among pregnant women in China, and this probability value of 0.2561 was used in the formula. This initial calculation resulted in a sample size of 293 participants. Considering the finite population, the sample size was further adjusted, yielding a target sample of 275 participants. Anticipating a potential 10% attrition rate, the final target sample size for this study was set at 303 participants.

### 2.3 Instruments

The instruments utilized in the study included: (1) sociodemographic information encompassing age, nationality, occupation, education level, monthly household income, number of live births, pre-pregnancy BMI, obstetric history, and PA characteristics; (2) the Chinese version of the Pregnancy Physical Activity Questionnaire (PPAQ), translated by Zhang et al. in 2013 [30]

from the original English version by Chasan-Taber et al. in 2004 [31]. The Chinese PPAQ consists of 31 items categorized into 4 domains: household/caregiving activities (13 items), occupational activities (5 items), exercise activities (8 items), and transportation/travel activities (5 items). Activities are classified by metabolic equivalent (MET) values as sedentary, light, moderate, or vigorous intensity. Each activity offers 6 frequency and duration options with corresponding weighting coefficients to calculate total and average daily energy expenditure [32]. The Chinese PPAQ has shown good face and expert content validity (0.940), along with strong test-retest reliability (intraclass correlation coefficient: 0.944 for total activity, 0.961 for light-intensity activity, 0.877 for moderate-intensity activity, 1.000 for vigorous-intensity activity, and 0.911 for sedentary activity), indicating its reliability and validity for assessing PA during pregnancy in the Chinese population.

## 2.4 Data collection

To recruit participants, the researchers initially uploaded the informed consent form and questionnaire to an online survey platform. Subsequently, they identified pregnancy-related social media groups in Macao, secured permission from group administrators to share the survey link, and requested them to distribute the link among group members. The informed consent form was displayed on the first page of the questionnaire, requiring participants to indicate their agreement before proceeding with the survey.

## 2.5 Data analysis

The data analysis was performed using IBM SPSS Statistics (version 29, IBM SPSS). Descriptive statistics, including frequencies and percentages, were used to illustrate the sociodemographic and obstetric histories, as well as the exercise habits, of the study participants.

The total energy expenditure (TEE) during physical activities of varying intensities and types, the proportions of TEE at different intensities and types, and the corresponding means, medians, and interquartile ranges (IQRs) were calculated. The energy expenditure for different intensities and types of PA exhibited a skewed distribution. Kruskal-Wallis tests and chi-square tests were employed to assess the statistical differences in PA levels across different trimesters of pregnancy. Chi-square tests were also used to evaluate the statistical differences in the proportions of women in different PA categories (sedentary and active) based on sociodemographic and obstetric characteristics. Multivariable binary logistic regression analyses were performed to compute odds ratios (ORs) and 95% confidence intervals (CIs). All statistical tests were two-sided, and a p-value less than 0.05 was considered statistically significant.

## 2.6 Ethical considerations

A written research protocol has been reviewed and approved by the Ethics Review Committee of Macao Polytechnic University (RP/AE-02/2022/E01). Before accessing the online survey, participants will be presented with an electronic informed consent document. Participants must provide their agreement to participate before proceeding to the survey. The study was conducted in full compliance with the principles of confidentiality, honesty, beneficence, and non-maleficence.

## 3. Results

The study collected a total of 314 questionnaires. Among these, 1 had a total energy consumption of 0 MET-hours/d, 5 had duplicate IP addresses, and 2 had illogical height and weight,

which were considered invalid and excluded. The final analysis included 306 valid questionnaires, resulting in an effective response rate of 97.45%.

### 3.1 Participants' sociodemographic characteristics

Among the 306 pregnant women, the majority (77.8%) were aged between 26–35 years old, 80.7% were still employed during their pregnancy, and 88.9% had a college degree or higher. The majority (84.3%) were in their second or third trimester, with 40.2% being first-time mothers and 59.8% having more than one child. Additionally, 42.2% had no exercise habits before pregnancy, and only 15% reported exercising regularly or daily (Table 1).

### 3.2 Participants' PA characteristics across pregnancy stages

In terms of activity intensity, the 306 pregnant women had an average weekly total energy expenditure of 202.82 MET-hours, with 187.93 MET-hours in early pregnancy, 185.21 MET-hours in mid-pregnancy, and 217.35 MET-hours in late pregnancy. The time spent on sedentary activities was 7.8%, low-intensity activities 51.6%, moderate-intensity activities 30.0%, and high-intensity activities 10.6%. Regarding activity types, household chores accounted for 33.9%, occupational activities 36.2%, transportation 20.8%, and exercise less than 3% (Table 1). Among the 306 pregnant women, 79.2%, 86.6%, and 87.6% met the threshold of >7.5 MET-hours/week of moderate-intensity activity in early, mid, and late pregnancy, respectively (Table 2). Late pregnancy had a higher energy expenditure on low-intensity activities compared to early and mid-pregnancy (P<0.05), while the differences in other pregnancy stages were not statistically significant (P>0.05) (Table 2).

### 3.3 Bivariate analysis of participants' PA status

Pregnant women were divided into a sedentary group (n = 43) and an active group (n = 263) based on whether they met the threshold of >7.5 MET-hours/week of moderate-intensity activity [1]. The results showed statistically significant differences between the two groups in employment status during pregnancy and the number of children (P<0.05). The proportion of employed pregnant women meeting the moderate-intensity activity threshold (82.5%) was higher than those who were not employed (67.4%). Primiparous women (62.7%) were less likely to meet the moderate-intensity activity threshold compared to multiparous women (37.3%). The differences in other variables were not statistically significant (P>0.05) (Table 3).

### 3.4 Multivariate analysis of participants' PA status

Using the achievement of the moderate-intensity activity threshold as the dependent variable (0 = sedentary group, 1 = active group), a forward conditional binary logistic regression analysis was performed. The results showed that being employed during pregnancy was a protective factor, as employed pregnant women had a 57.9% lower risk of being sedentary compared to those who were not employed (OR = 0.421, 95%CI:0.196–0.902). Additionally, multiparous women with two or more children had an 8.090 times higher chance of meeting the moderate-intensity activity threshold compared to nulliparous women (OR = 9.090, 95%CI:2.071–39.905) (Table 4), and these two factors explained 85.9% of the variance

## 4. Discussion

### 4.1 Sociodemographic characteristics

The present study surveyed a total of 306 pregnant women, with the majority (77.8%) aged between 26–35 years old. In terms of employment status, only 19.3% of the pregnant women

**Table 1. Socio-demographic characteristics and health factors of study participants in Macao (n = 306).**

| Variables | N(%) |
|---|---:|
| Age group(years) | |
| ≤25 | 14(4.6) |
| 26–30 | 101(33.0) |
| 31–35 | 137(44.8) |
| ≥36 | 54(17.6) |
| Nationality | |
| China | 299(97.7) |
| Others | 7(2.3) |
| Working during pregnancy | |
| Yes | 247(80.7) |
| No | 59(19.3) |
| Level of education | |
| High school and below | 34(11.1) |
| College or university | 214(69.9) |
| Postgraduate and above | 58(19.0) |
| Household monthly income (MOP) | |
| ≤30,000 | 142(46.4) |
| 30,001–50,000 | 94(30.7) |
| 50,001–70,000 | 41(13.4) |
| ≥70,001 | 29(9.5) |
| Stage of pregnancy | |
| First trimester ≤13 weeks | 48(15.7) |
| Second trimester ≥14, <28 weeks | 97(31.7) |
| Third trimester ≥28 weeks | 161(52.6) |
| Parity at recruitment | |
| 0 child | 123(40.2) |
| 1 child | 120(39.2) |
| ≥2 children | 63(20.6) |
| Pre-pregnancy BMI | |
| Underweight: < 18.5 kg/m2 | 34(11.1) |
| Normal: 18.5–24.99 kg/m2 | 164(53.6) |
| Overweight: 25–29.99 kg/m2 | 52(17.0) |
| Obesity: ≥30 kg/m2 | 56(18.3) |
| Exercise habits before pregnancy | |
| Never do any exercise (less than once a month) | 129(42.2) |
| Occasionally do exercise (about 1–2 times a month) | 87(28.4) |
| Sometimes do exercise (about 1–2 times every half month) | 44(14.4) |
| Do regular exercise (about 1–2 times a week) | 38(12.4) |
| Exercise almost every day (about 5 times a week or more) | 8(2.6) |

were unemployed. This is in contrast to a nearby region, Hong Kong, where 36.8% of pregnant women were full-time homemakers, indicating a higher employment rate among pregnant women in Macao [24]. Additionally, the study participants had relatively high educational levels, with nearly 90% holding a college or university degree. As a developed region, Macao has quality educational resources and a 15-year compulsory education policy, with a steadily increasing university enrollment rate among high school graduates, rising from 76.8% two

**Table 2. Physical activities of study participants across trimesters in Macao [Mean, median (IQR), MET-hours/ week, n = 306].**

| Physical activities (MET-h/week) | Total (n = 306) | 1ST trimester ≤13 weeks (n = 48) | 2nd trimester ≥14, <28 weeks (n = 97) | 3rd trimester ≥28 weeks (n = 161) | H (Kruskal-Wallis)/ $\chi^2$ | P |
|---|---|---|---|---|---|---|
| Total energy expenditure per week | 202.83,158.91 (148.36) | 187.93,155.18(168.99) | 185.21,155.88(107.48) | 217.35,161.35(158.97) | 4.838 | 0.089 |
| By activity intensity (n = 306) | | | | | | |
| Sedentary | 15.82,12.60(14.41) | 14.53,10.85(14.88) | 14.63,14.00(14.35) | 16.93,14.70(14.57) | 1.739 | 0.419 |
| Light | 104.56,88.36 (78.63) | 93.66,77.26(85.32) | 91.60,79.80(69.61) | 115.09,99.54(83.76) | 7.951 | 0.019* |
| Moderate | 60.92,28.94(66.33) | 63.38,31.58(95.9) | 54.42,26.56(53.39) | 64.10,30.55(70.40) | 1.471 | 0.479 |
| Vigorous | 21.52,7.50(21.28) | 16.36,7.53(21.28) | 24.56,7.56(21.70) | 21.24,6.72(21.28) | 1.215 | 0.545 |
| By type of activity | | | | | | |
| Household/caregiving | 68.68,44.15(75.86) | 69.53,44.70(83.08) | 58.66,35.76(62.36) | 74.46,50.62(79.21) | 3.294 | 0.193 |
| Occupational (n = 247) [a] | 73.35,66.15(56.07) | 71.25,67.20(50.57) | 67.43,57.42(47.42) | 77.41,69.05(52.44) | 1.385 | 0.500 |
| Sports/exercise | 5.76,2.56(4.71) | 6.91,3.33(57.85) | 5.08,2.56(4.80) | 5.83,2.56(5.33) | 0.617 | 0.734 |
| Transportation | 42.26,24.08(42.56) | 34.60,21.39(37.38) | 42.12,24.08(31.92) | 44.62,29.33(41.93) | 1.902 | 0.386 |
| Inactivity | 24.24,19.95(19.72) | 20.49,16.68(19.04) | 23.03,19.95(20.14) | 26.09,21.00(26.31) | 2.374 | 0.305 |
| Meet the guideline, n(%) | 263(85.9) | 38(79.2) | 84(86.6) | 141(87.6) | 2.216 | 0.330 |

a for occupational activity: 1st trimester = 38, 2nd trimester = 76, and 3rd trimester = 133; * for P value<0.05

decades ago to the current 95.2% [33], contributing to the high educational attainment of the pregnant women in this study.

## 4.2 PA at different intensities

The results of this research show that Macao's pregnant women spend nearly 60% of their weekly time in sedentary and low-intensity physical activities, with high-intensity physical activities accounting for only about 10%. Moderate-intensity physical activities comprise approximately 30% of the total energy expenditure throughout the pregnancy period, similar to the findings in Hong Kong (23%-30%) [24], but higher than in mainland China (21.2%-24.4%) [34]. According to the standard of moderate-intensity PA energy expenditure > 7.5 MET-hours/week [1], 85.9% of the participants in this study met the criteria, which is higher than the results for white British women (61%) [35], mainland China (18.07%) [36], and Ireland (21.5%) [37], but lower than a study on pregnant women in the United States (100%) [38].

## 4.3 Engagement in different types of activities

Regarding the types of activities, the highest proportion of energy expenditure was from occupational activities (36.2%), followed by household chores (33.9%). Transportation also accounted for approximately one-fifth of the total energy expenditure, while less than 3% was spent on exercise/training, which is similar to a Canadian study [39]. In contrast, a study in mainland China showed a higher proportion of energy expenditure from household chores (48%), followed by occupational activities (40%), with the proportion spent on exercise/training being consistent with the present study (less than 3%) [34]. A Brazilian study also indicated that the highest proportion of energy expenditure among pregnant women was in the domestic domain, including household chores and caring for children, the elderly, or pets [40]. Regardless of the level of economic development, women are traditionally perceived as the primary caregivers in the family, responsible for a significant amount of domestic labor.

**Table 3. Bivariate analysis of factors associated with physical activity status among study participants in Macao (n = 306).**

| Variables | Sedentary (43) n (%) | Active (263) n (%) | $\chi^2$ | P |
|---|---|---|---|---|
| Age group(years) | | | | |
| ≤25 | 1(2.3) | 13(4.9) | 2.549 | 0.466 |
| 26–30 | 13(30.2) | 88(33.5) | | |
| 31–35 | 18(41.9) | 119(45.2) | | |
| ≥36 | 11(25.6) | 43(16.3) | | |
| Nationality | | | | |
| China | 41(95.3) | 258(98.1) | 1.250 | 0.263 |
| Others | 2(4.7) | 5(1.9) | | |
| Working during pregnancy | | | | |
| Yes | 29(67.4) | 217(82.5) | 5.323 | 0.021* |
| No | 14(32.6) | 46(17.5) | | |
| Level of education | | | | |
| High school and below | 2(4.7) | 32(12.2) | 2.401 | 0.301 |
| College or university | 31(72.1) | 183(69.6) | | |
| Postgraduate and above | 10(23.3) | 48(18.3) | | |
| Household monthly income (MOP) | | | | |
| ≤30,000 | 21(48.8) | 121(46.0) | 1.525 | 0.677 |
| 30,001–50,000 | 10(23.3) | 84(31.9) | | |
| 50,001–70,000 | 7(16.3) | 34(12.9) | | |
| ≥70,001 | 5(11.6) | 24(9.1) | | |
| Stage of pregnancy | | | | |
| First trimester ≤13 weeks | 10(23.3) | 38(14.4) | 2.216 | 0.330 |
| Second trimester ≥14, <28 weeks | 13(30.2) | 84(31.9) | | |
| Third trimester ≥28 weeks | 20(46.5) | 141(53.6) | | |
| Parity at recruitment | | | | |
| 0 child | 27(62.8) | 96(36.5) | 13.092 | 0.001* |
| 1 child | 14(32.6) | 106(40.3) | | |
| ≥2 children | 2(4.7) | 61(23.2) | | |
| Pre-pregnancy BMI | | | | |
| Underweight: < 18.5 kg/m2 | 5(11.6) | 29(11.0) | 0.589 | 0.899 |
| Normal: 18.5–24.99 kg/m2 | 25(58.1) | 139(52.9) | | |
| Overweight: 25–29.99 kg/m2 | 6(14.0) | 46(17.5) | | |
| Obesity: ≥30 kg/m2 | 7(16.3) | 49(18.6) | | |
| Exercise habits before pregnancy | | | | |
| Never do any exercise (less than once a month) | 14(32.6) | 115(43.7) | 2.408 | 0.661 |
| Occasionally do exercise (about 1–2 times a month) | 14(32.6) | 73(27.8) | | |
| Sometimes do exercise (about 1–2 times every half month) | 7(16.3) | 37(14.1) | | |
| Do regular exercise (about 1–2 times a week) | 6(14.0) | 32(12.2) | | |
| Exercise almost every day (about 5 times a week or more) | 2(4.7) | 6(2.3) | | |

The present study revealed that the highest proportion of energy expenditure among pregnant women was from occupational activities, which may be related to the fact that nearly 81% of the participants were still employed. This reflects the modern societal trend where pregnant women continue to actively participate in professional work to maintain their economic income and social status. The relatively high proportion of energy expenditure from transportation activities may be associated with the level of urbanization and the need for pregnant women to commute to work during their pregnancy. The higher frequency and intensity of

**Table 4. Multivariate analysis of factors associated with physical activity status among study participants in Macao (n = 306).**

| Variables | B | P | OR | 95%C.I. for OR | |
|---|---|---|---|---|---|
| | | | | Lower | Upper |
| Working during pregnancy | -0.865 | 0.026 | 0.421 | 0.196 | 0.902 |
| Parity at recruitment:0 child | | 0.002 | | | |
| Parity at recruitment: 1 child | 0.877 | 0.017 | | | |
| Parity at recruitment: ≥2 children | 2.207 | 0.003 | 9.090 | 2.071 | 39.905 |
| Constant | 2.286 | <0.001 | 9.831 | | |

travel among pregnant women contribute to a larger share of energy expenditure from transportation. Consistently, the proportion of energy expenditure from exercise/training was very low across all studies (citations), which may be attributed to pregnant women's concerns about the potential impact of exercise on fetal health, as well as their susceptibility to fatigue due to physiological changes.

### 4.4 PA energy expenditure

The study showed the median weekly energy expenditure of women in early and mid-pregnancy was 155.18 MET-h/W and 155.88 MET-h/W, respectively, which was slightly lower than that of pregnant women in the nearby region of Hong Kong (176.6 MET-h/W and 179.4 MET-h/W) [24], but higher than that of pregnant women in Japan (137.9 MET-h/W and 151.9 MET-h/W) [41]. This was significantly higher than the energy expenditure of pregnant women in Ethiopia (3.23 MET-h/W and 3.35 MET-h/W) [42]. When comparing the mean values, the weekly energy expenditure of women in early, mid, and late pregnancy in the current study was 187.93 MET-h/W, 185.21 MET-h/W, and 217.35 MET-h/W, respectively. This was lower than that of pregnant women in Portugal (270.915 MET-h/W, 220.541 MET-h/W, and 210.348 MET-h/W) [43], but higher than that of pregnant women in mainland China (182.0 MET-h/W, 168.8 MET-h/W, and 157.4 MET-h/W) [34]. It is evident that some studies conducted in mainland China and other countries have shown a decreasing trend in the total PA of pregnant women from early to mid to late pregnancy [34, 43]. However, the present study found that the energy expenditure from PA was similar between early and mid-pregnancy, and increased in late pregnancy, although low-intensity activities predominated during late pregnancy (P<0.05).

The overall weekly energy expenditure of the pregnant women in this study fell within the range between the values reported for pregnant women in the nearby region of Hong Kong and Japan, and was higher than that of the remote region of Ethiopia. This may reflect the influence of geographical location, economic development level, and lifestyle factors on the energy expenditure of pregnant women. Pregnant women in relatively affluent regions tend to have more resources invested in health management and daily activities. Compared to pregnant women in Portugal, the participants in the current study had slightly lower weekly energy expenditure, which may be related to differences in cultural background and social expectations. Portuguese pregnant women may emphasize PA more [43], while Chinese pregnant women tend to focus on pregnancy care, leading to relatively lower energy expenditure. Compared to studies conducted in other regions of China, the present study found an increase in energy expenditure during late pregnancy, whereas other studies have generally shown a gradual decreasing trend. This may be related to the lifestyle and care practices of the pregnant women in the study area. Some late-pregnancy household chores and transportation activities may have increased, offsetting the impact of decreased PA capacity.

#### 4.5 Factors influencing SB during pregnancy

This study found that continued employment during pregnancy was a protective factor against SB, with employed pregnant women having a 57.9% lower likelihood of failing to meet PA guidelines compared to their unemployed counterparts (OR = 0.421, 95%CI:0.196–0.902), consistent with findings from previous international research [42]. Workplaces typically require a certain degree of PA, such as walking and standing, to complete job tasks. In contrast, unemployed pregnant women may be more prone to a sedentary lifestyle, with relatively less PA. In addition, employment often involves commuting to the workplace, which typically entails walking, driving, or using other modes of transportation, thereby increasing daily energy expenditure. Conversely, unemployed pregnant women who remain at home have relatively less commuting-related PA, and thus lower energy expenditure.

This study also found that multiparous women with two or more children were 9.090 times more likely to meet moderate-intensity PA guidelines compared to nulliparous women (OR = 9.090, 95%CI:2.071–39.905), consistent with many previous studies [42, 44]. This likely attribute to the fact that primiparous women may have less experience and be more concerned about the physical changes and activity requirements during pregnancy, and thus are more hesitant to engage in strenuous PA, fearing potential adverse effects on the fetus. Multiparous women, on the other hand, may have a better understanding and more confidence regarding this aspect. Primiparous women's focus is often more centered on their own and the fetus's care, while multiparous women also need to manage the care of their other children and perform various household chores. These daily living and caregiving demands may increase the PA levels of multiparous women.

The research findings indicate that primiparous women and unemployed pregnant women are more likely to fail to meet the recommended energy expenditure targets compared to working women and those with multiple children. This suggests that nursing practice should particularly focus on these vulnerable groups by providing targeted prenatal education and home visits to offer tailored nutritional guidance and lifestyle recommendations to help them maintain a healthy energy balance. Nursing interventions should also involve facilitating the establishment of support networks for these women, who tend to lack the social interactions afforded by the work environment and peer groups, by encouraging participation in prenatal classes and pregnancy support groups to meet their emotional and practical needs.

This study did not find any association between educational level and inadequate PA during pregnancy, consistent with findings from Italy [23] and Mainland China [14], but in contrast with a study from Hong Kong [24]. This may be related to the fact that nearly 90% of the participants in this study had a relatively high educational level. This study also did not find any associations between age, household monthly income, prenatal exercise habits, and pre-pregnancy BMI with inadequate PA during pregnancy. Future research should continue to explore these relationships, potentially by expanding the sample size and using random sampling methods to better represent the study population.

#### 4.6 Limitations

The present study represents the first investigation into the PA and SB of pregnant women in Macao. However, several limitations need to be considered. Firstly, the data were limited to self-reported information from participants, and recall bias may be present due to the complex nature of PA and SB. Secondly, as a result of convenient sampling, the generalizability of the study population may be limited. Future research should include the assessment of activity using objective methods, such as pedometers or accelerometers, as well as the collection of data throughout the entire pregnancy period. Additionally, this study employs a cross-

sectional design, which does not allow for the establishment of causal relationships between the identified factors and physical activity among pregnant women. Future studies could consider utilizing cohort or experimental research designs to address these issues.

## 5. Conclusion

This pioneering study on pregnant women in Macao reveals a relatively low prevalence of SB. Employed and multiparous women are more prone to meet activity recommendations. Enhanced education is crucial, especially for primiparous and unemployed pregnant women, to reduce SB and promote healthier habits.

## Supporting information

**S1 Data.**
(XLSX)

## Acknowledgments

The authors express their sincere appreciation to all participants for their involvement in this study.

## Author Contributions

**Conceptualization:** Xin Wang.

**Data curation:** Ka Chon Mok.

**Formal analysis:** Ka Chon Mok, Xin Wang.

**Funding acquisition:** Xin Wang.

**Investigation:** Ka Chon Mok.

**Methodology:** Xin Wang.

**Project administration:** Xin Wang.

**Supervision:** Ming Liu.

**Writing – original draft:** Ka Chon Mok, Xin Wang.

**Writing – review & editing:** Ming Liu, Xin Wang.

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
