## [Decision Letter · Decision Letter 0]

19 Nov 2024

PONE-D-24-37343What are the contributing factors to sedentary behavior among pregnant women? A cross-sectional studyPLOS ONE

Dear Dr. Wang,

Thank you for submitting your manuscript to PLOS ONE. After careful consideration, we feel that it has merit but does not fully meet PLOS ONE’s publication criteria as it currently stands. Therefore, we invite you to submit a revised version of the manuscript that addresses the points raised during the review process.

ACADEMIC EDITOR: Dear Author, please attend to all comments provided by the reviewers and make necesssary corrections.

We look forward to receiving your revised manuscript.

Kind regards,

Zulkarnain Jaafar

Academic Editor

PLOS ONE

Journal Requirements: When submitting your revision, we need you to address these additional requirements. 1. Please ensure that your manuscript meets PLOS ONE's style requirements, including those for file naming. The PLOS ONE style templates can be found at https://journals.plos.org/plosone/s/file?id=wjVg/PLOSOne_formatting_sample_main_body.pdf and https://journals.plos.org/plosone/s/file?id=ba62/PLOSOne_formatting_sample_title_authors_affiliations.pdf 2. Thank you for stating the following financial disclosure: "This work was supported by Macao Polytechnic University [grant number RP/AE-02/2022]" Please state what role the funders took in the study.  If the funders had no role, please state: ""The funders had no role in study design, data collection and analysis, decision to publish, or preparation of the manuscript."" If this statement is not correct you must amend it as needed. Please include this amended Role of Funder statement in your cover letter; we will change the online submission form on your behalf. 3. We are unable to open your Supporting Information file "PA&SB data.sav". Please kindly revise as necessary and re-upload. 4. We note that your Data Availability Statement is currently as follows: All relevant data are within the manuscript and its Supporting Information files. Please confirm at this time whether or not your submission contains all raw data required to replicate the results of your study. Authors must share the “minimal data set” for their submission. PLOS defines the minimal data set to consist of the data required to replicate all study findings reported in the article, as well as related metadata and methods (https://journals.plos.org/plosone/s/data-availability#loc-minimal-data-set-definition). For example, authors should submit the following data: - The values behind the means, standard deviations and other measures reported;- The values used to build graphs;- The points extracted from images for analysis. Authors do not need to submit their entire data set if only a portion of the data was used in the reported study. If your submission does not contain these data, please either upload them as Supporting Information files or deposit them to a stable, public repository and provide us with the relevant URLs, DOIs, or accession numbers. For a list of recommended repositories, please see https://journals.plos.org/plosone/s/recommended-repositories. If there are ethical or legal restrictions on sharing a de-identified data set, please explain them in detail (e.g., data contain potentially sensitive information, data are owned by a third-party organization, etc.) and who has imposed them (e.g., an ethics committee). Please also provide contact information for a data access committee, ethics committee, or other institutional body to which data requests may be sent. If data are owned by a third party, please indicate how others may request data access. 5. Please ensure that you include a title page within your main document. You should list all authors and all affiliations as per our author instructions and clearly indicate the corresponding author. 6. Please include your full ethics statement in the ‘Methods’ section of your manuscript file. In your statement, please include the full name of the IRB or ethics committee who approved or waived your study, as well as whether or not you obtained informed written or verbal consent. If consent was waived for your study, please include this information in your statement as well. 7. Please include captions for your Supporting Information files at the end of your manuscript, and update any in-text citations to match accordingly. Please see our Supporting Information guidelines for more information: http://journals.plos.org/plosone/s/supporting-information.

Reviewers' comments:

Reviewer's Responses to Questions

**Comments to the Author**

1. Is the manuscript technically sound, and do the data support the conclusions?

Reviewer #1: Yes

Reviewer #2: Partly

2. Has the statistical analysis been performed appropriately and rigorously? 

Reviewer #1: Yes

Reviewer #2: Yes

3. Have the authors made all data underlying the findings in their manuscript fully available?

Reviewer #1: Yes

Reviewer #2: Yes

4. Is the manuscript presented in an intelligible fashion and written in standard English?

Reviewer #1: No

Reviewer #2: Yes

5. Review Comments to the Author

Reviewer #1: I applaud the authors of this article for their efforts to came out this interesting findings. I have appended some comments and suggestions to the authors. Please, consider my comments depicted hereunder to boost the scientific quality of your articles.

Title: What are the contributing factors to sedentary behaviour among pregnant women? A cross-sectional study

It seems like a specific research question rather than title.

The title should express the study setting

Abstract:

Authors would follow the Journal’s guideline. I think PLOS ONE journal requires structured abstract.

Introduction

Line no. 27-30: the sentence is too long and incomplete.

Line no. 51-53: The statement is not clear. Maternal age and gravidity are different characteristics.

This section did not state the regional and local evidence about SB. For instance, the prevalence of SB in china would be stated.

You would also state the application of the World Health Organization’ updated guidelines on lifetime PA in china and or your study area. In addition, any measures (strategies) undertaken by the local government to promote PA among your study population would be mentioned.

You mentioned that the habitants of Macao has unique blend of Chinese and Western cultures. So you would use a qualitative or mixed study design to explore potentially cultural- and tradition-related attributes of SB.

Materials and methods

You would describe the study setting in detail.

Your study’s period was from October 1, 2023, to February 29, 2024. The estimated number of pregnant women of the same period in 2022 would be used. The whole 2022 data might not be valid for this specific period because the number of pregnant women could not be uniformly distributed in different months per year.

“proficient in reading and writing Chinese”. It seems discrimination. Language translation could solve this bias.

Sampling technique and procedure, study’s outcome, operational definitions, and data quality control measures were not stated.

Model assumptions and selection strategies would be stated.

Result

Ensure that the reported findings in line with the predetermined objective.

Discussion

Ensure that the discussion in line with the predetermined objective.

Reviewer #2: This study examines the relationship between the prevalence of sitting behavior among pregnant women in Macau and their sociodemographic attributes. Still, some points need to be questioned or corrected.

1. Title

The title could be more concise by focusing on "Sedentary Behavior Among Pregnant Women in Macau" or similar. The authors should mention Macau's regional context.

2. Novelty

In lines 59-60, the authors state that the factors that affect PA have not been fully elucidated in previous research. The purpose of this research should be to fill this research gap. It is necessary to clearly state how the authors resolved the inadequacy and difficulties of previous studies and the novelty of this research. If the regional context of Macau is the core of the novelty, this should be clearly stated in the title and purpose. It may also be worth noting any potential regional limitations or biases related to the Macao setting, such as healthcare access or support systems.

Cross-sectional, descriptive research is insufficient for elucidating causal relationships. The limitations of this study in elucidating the factors that affect PA should also be mentioned in 4.6 Limitations.

3. Sample

Please add the assumptions used by the authors when estimating the number of pregnant women from the number of newborns.

Please explain the sample size calculation method in detail by indicating the assumed effect size and statistical analysis method.

Please add specifics about the criteria used to select social media groups (e.g., size or activity level) that could provide additional clarity.

4. Analysis

Please show the specific analysis methods used for multivariate analysis by describing the coefficient of determination and effect size and the appropriateness of the sample size.

5. Style

In Table 2, please correct "3rd Third". The table should be modified to reflect the APA style.

6. PLOS authors have the option to publish the peer review history of their article (what does this mean?). If published, this will include your full peer review and any attached files.

Reviewer #1: **Yes: **Wubishet Gezimu

Reviewer #2: No

---

## [Author Response · Author response to Decision Letter 0]

2 Jan 2025

Journal Requirements:

Response: The format of the manuscript has been checked and revised according to PLOS ONE's style requirements.

"This work was supported by Macao Polytechnic University [grant number: RP/AE-02/2022]"

Response: The amended Role of Funder statement was added in the cover letter: "The funders had no role in study design, data collection and analysis, decision to publish, or preparation of the manuscript."

3. We are unable to open your Supporting Information file "PA&SB data.sav". Please kindly revise as necessary and re-upload.

Response: The file “PA&SB data.sav” is an SPSS data file that must be opened with SPSS software. To facilitate your review, I have converted this data file to Excel format. However, please note that some categorical data values may not be displayed correctly in this format. I appreciate your understanding. I will re-upload the data in Excel format.

Response: I confirm that my submission contains all original data necessary to replicate the results of this study and that all relevant data are included in the manuscript and its supporting information files.

However, these original data cannot be shared because the data contains potentially sensitive information, such as the address and contact information of the research subjects. The Ethics Committee of Macau Polytechnic University requires that the privacy of the research subjects be kept confidential, the researcher must not release personal identifier information to unauthorized personnel. 

The contact information of the Ethics Committee of Macao Polytechnic University is email: dei@mpu.edu.mo, phone: (853)85996333.

5. Please ensure that you include a title page within your main document. You should list all authors and all affiliations as per our author instructions and clearly indicate the corresponding author.

Response: I have included the authors information within the main manuscript, accordance with the formatting requirements of PLOS ONE.

Response: Revised, please see “Revised Manuscript with Track Changes” line no. 179-180.

Response: The supporting documents “PA&SB data” provided by the author are intended solely for review by the publisher's editors and external peer reviewers, and are not suitable for sharing. This is due to the requirements of the ethics committee approval. For more specific information, please refer to the response to question 4. Thank you for your understanding.

Review Comments to the Author

Reviewer #1: I applaud the authors of this article for their efforts to came out this interesting findings. I have appended some comments and suggestions to the authors. Please, consider my comments depicted hereunder to boost the scientific quality of your articles.

Title: What are the contributing factors to sedentary behaviour among pregnant women? A cross-sectional study

It seems like a specific research question rather than title.

The title should express the study setting

Response: The title is modified to “The physical activity and sedentary behavior among pregnant women in Macao: A cross-sectional study”.

Abstract:

Authors would follow the Journal’s guideline. I think PLOS ONE journal requires structured abstract.

Response: Revised, please see P1-2.

Introduction

Line no. 27-30: the sentence is too long and incomplete.

Response: Revised, please see “Revised Manuscript with Track Changes” line no. 43-46.

Line no. 51-53: The statement is not clear. Maternal age and gravidity are different characteristics.

Response: Revised, please see “Revised Manuscript with Track Changes” Line no.71-74.

This section did not state the regional and local evidence about SB. For instance, the prevalence of SB in china would be stated.

Response: Revised, please see “Revised Manuscript with Track Changes” Line no.85-87.

You would also state the application of the World Health Organization’ updated guidelines on lifetime PA in China and or your study area. In addition, any measures (strategies) undertaken by the local government to promote PA among your study population would be mentioned.

Response: Revised, please see “Revised Manuscript with Track Changes” Line no.88-93.

You mentioned that the habitants of Macao has unique blend of Chinese and Western cultures. So you would use a qualitative or mixed study design to explore potentially cultural- and tradition-related attributes of SB.

Response: Thank you for your suggestions. Future research will consider continuing with qualitative or mixed-methods studies.

Materials and methods

You would describe the study setting in detail.

Response: Revised, please see “Revised Manuscript with Track Changes” Line no.104-106.

Your study’s period was from October 1, 2023, to February 29, 2024. The estimated number of pregnant women of the same period in 2022 would be used. The whole 2022 data might not be valid for this specific period because the number of pregnant women could not be uniformly distributed in different months per year.

Response: As explained in lines 111 to 121 of the manuscript, we considered several factors in calculating the sample size. As you noted, the number of pregnant women cannot be evenly distributed across the different months of the year, and we cannot predict how long it will take to collect the required sample size. Therefore, we used the total number of newborns over an entire year to estimate the number of pregnant women. Our aim is to calculate the sample size based on a population estimate that is as close as possible to the actual number, ensuring representativeness for the overall population.

“proficient in reading and writing Chinese”. It seems discrimination. Language translation could solve this bias.

Response: The survey was collected via online media platforms (WeChat groups), and the entire questionnaire was available only in Chinese. Specifically, the questionnaire measuring physical activity among pregnant women utilized the Chinese version, which differs in terms of the number of questions and wording compared to the English version. Given that the target population for this study primarily consists of local pregnant women in Macao, and considering that researchers could not assess the accuracy of translations provided by participants' translation software, we included language requirements in the inclusion criteria. This decision was made to ensure clarity and understanding, and it should not be interpreted as "language discrimination."

Sampling technique and procedure, study’s outcome, operational definitions, and data quality control measures were not stated.

Response: The sampling technique and procedure was described in line no. 102-106, and the data quality control measures was stated in line no. 187-190. While, the operational definitions were described in line no. 222-223, 235-236.

Model assumptions and selection strategies would be stated.

Response: The model selection strategy can be found on line no.236-237.

Result

Ensure that the reported findings in line with the predetermined objective.

Response: yes.

Discussion

Ensure that the discussion in line with the predetermined objective.

Response: yes.

Reviewer #2: This study examines the relationship between the prevalence of sitting behavior among pregnant women in Macau and their sociodemographic attributes. Still, some points need to be questioned or corrected.

1. Title

The title could be more concise by focusing on "Sedentary Behavior Among Pregnant Women in Macau" or similar. The authors should mention Macau's regional context.

Response: The title is modified to “The physical activity and sedentary behavior among pregnant women in Macao: A cross-sectional study”.

2. Novelty

In lines 59-60, the authors state that the factors that affect PA have not been fully elucidated in previous research. The purpose of this research should be to fill this research gap. It is necessary to clearly state how the authors resolved the inadequacy and difficulties of previous studies and the novelty of this research. If the regional context of Macau is the core of the novelty, this should be clearly stated in the title and purpose. It may also be worth noting any potential regional limitations or biases related to the Macao setting, such as healthcare access or support systems.

Response: Revised, please see “Revised Manuscript with Track Changes” Line no.90-93.

Cross-sectional, descriptive research is insufficient for elucidating causal relationships. The limitations of this study in elucidating the factors that affect PA should also be mentioned in 4.6 Limitations.

Response: Revised, please see “Revised Manuscript with Track Changes” Line no.381-384.

3. Sample

Please add the assumptions used by the authors when estimating the number of pregnant women from the number of newborns.

Response: Revised, please see “Revised Manuscript with Track Changes” Line no.113-116.

Please explain the sample size calculation method in detail by indicating the assumed effect size and statistical analysis method.

Response: The sample size calculation method is detailed in the manuscript, please see Line no.130-136.

Please add specifics about the criteria used to select social media groups (e.g., size or activity level) that could provide additional clarity.

Response: Revised, please see Line no.104-106.

4. Analysis

Please show the specific analysis methods used for multivariate analysis by describing the coefficient of determination and effect size and the appropriateness of the sample size.

Response: Revised, please see Line no.242(This study used logistic regression analysis, not linear regression analysis, so no coefficient of determination and effect size were provided. However, we increased the percentage of explanations of the dependent variables by the variables screened out by the logistic regression analysis.).

5. Style

In Table 2, please correct "3rd Third". The table should be modified to reflect the APA style.

Response: Revised.

6. PLOS authors have the option to publish the peer review history of their article (what does this mean?). If published, this will include your full peer review and any attached files.

Do you want your identity to be public for this peer review? For information about this choice, including consent withdrawal, please see our Privacy Policy.

Reviewer #1: Yes: Wubishet Gezimu

Reviewer #2: No

---

## [Decision Letter · Decision Letter 1]

15 Jan 2025

The physical activity and sedentary behavior among pregnant women in Macao: A cross-sectional study

PONE-D-24-37343R1

Dear Dr. XIN WANG,

We’re pleased to inform you that your manuscript has been judged scientifically suitable for publication and will be formally accepted for publication once it meets all outstanding technical requirements.

Kind regards,

Zulkarnain Jaafar

Academic Editor

PLOS ONE

Additional Editor Comments (optional):

Reviewers' comments:

Reviewer's Responses to Questions

**Comments to the Author**

1. If the authors have adequately addressed your comments raised in a previous round of review and you feel that this manuscript is now acceptable for publication, you may indicate that here to bypass the “Comments to the Author” section, enter your conflict of interest statement in the “Confidential to Editor” section, and submit your "Accept" recommendation.

Reviewer #2: All comments have been addressed

2. Is the manuscript technically sound, and do the data support the conclusions?

Reviewer #2: Yes

3. Has the statistical analysis been performed appropriately and rigorously? 

Reviewer #2: Yes

4. Have the authors made all data underlying the findings in their manuscript fully available?

Reviewer #2: Yes

5. Is the manuscript presented in an intelligible fashion and written in standard English?

Reviewer #2: Yes

6. Review Comments to the Author

Reviewer #2: (No Response)

7. PLOS authors have the option to publish the peer review history of their article (what does this mean?). If published, this will include your full peer review and any attached files.

Reviewer #2: No

---

## [Editor Report · Acceptance letter]

22 Jan 2025

PONE-D-24-37343R1 

PLOS ONE

Dear Dr. WANG, 

I'm pleased to inform you that your manuscript has been deemed suitable for publication in PLOS ONE. Congratulations! Your manuscript is now being handed over to our production team.

Kind regards, 

on behalf of

Dr. Zulkarnain Jaafar 

Academic Editor

PLOS ONE